# H2A-H2B Histone Dimer Plasticity and Its Functional Implications

**DOI:** 10.3390/cells11182837

**Published:** 2022-09-12

**Authors:** Anastasiia S. Kniazeva, Grigoriy A. Armeev, Alexey K. Shaytan

**Affiliations:** Department of Biology, Lomonosov Moscow State University, 119234 Moscow, Russia

**Keywords:** nucleosome, chromatin, molecular modeling, histones, H2A-H2B dimers, MD simulations, structural bioinformatics, nucleosome sliding, histone variants

## Abstract

The protein core of the nucleosome is composed of an H3-H4 histone tetramer and two H2A-H2B histone dimers. The tetramer organizes the central 60 DNA bp, while H2A-H2B dimers lock the flanking DNA segments. Being positioned at the sides of the nucleosome, H2A-H2B dimers stabilize the overall structure of the nucleosome and modulate its dynamics, such as DNA unwrapping, sliding, etc. Such modulation at the epigenetic level is achieved through post-translational modifications and the incorporation of histone variants. However, the detailed connection between the sequence of H2A-H2B histones and their structure, dynamics and implications for nucleosome functioning remains elusive. In this work, we present a detailed study of H2A-H2B dimer dynamics in the free form and in the context of nucleosomes via atomistic molecular dynamics simulations (based on *X. laevis* histones). We supplement simulation results by comparative analysis of information in the structural databases. Particularly, we describe a major dynamical mode corresponding to the bending movement of the longest H2A and H2B α-helices. This overall bending dynamics of the H2A-H2B dimer were found to be modulated by its interactions with DNA, H3-H4 tetramer, the presence of DNA twist-defects with nucleosomal DNA and the amino acid sequence of histones. Taken together, our results shed new light on the dynamical mechanisms of nucleosome functioning, such as nucleosome sliding, DNA-unwrapping and their epigenetic modulation.

## 1. Introduction

The key element of eukaryotic chromatin is the nucleosome, a segment of around 200 DNA base pairs interacting with an octamer of histone proteins (H3, H4, H2A, H2B—two copies of each type) [1]. The octamer is tightly wrapped by a left-handed DNA superhelix, 145–147 base pairs in length, forming the nucleosome core particle (NCP) [2]. The histone octamer has a characteristic tripartite structure consisting of an H3-H4 tetramer, which interacts with the central 60 DNA base pairs, and two H2A-H2B dimers, which organize the remaining distal and proximal segments of the nucleosomal DNA. The nucleosomal DNA interacts with the octamer at 14 binding sites, and at every binding site, an arginine side chain is inserted into the DNA minor groove (Figure 1a). Each H2A-H2B dimer provides three binding sites: the central α1-α1 binding site with H2B R33 inserted into the DNA minor groove and two flanking L1L2 binding sites with H2A R42 and H2A R77 serving as minor groove anchors (Figure 1b–d).

Nucleosomes are now regarded as dynamic entities whose dynamics are essential for genome functioning. Modulation of these dynamics by histone post-translational modifications, histone variants, and interactions with chromatin proteins is an essential mechanism of epigenetic regulation of gene expression [3]. Dysregulation of nucleosome dynamics through mutations or aberrant histone post-translational modifications is associated with human disease, particularly cancer [4,5]. Such dynamical modes as DNA wrapping/unwrapping, nucleosome sliding, H2A-H2B dimer exchange are known to affect transcription factor binding [6,7], transcription by Pol II through nucleosomes [8], higher-order chromatin architecture [9], etc. Recently, however, it has been also shown that subtle dynamical changes within the histone octamer or even individual histone dimers have functional importance. For instance, the introduction of cross-links within the histone octamer affects nucleosome sliding [10], nucleosome remodeling [11] and the formation of higher-order chromatin structure [12]. Evidence exists for an allosteric communication through the histone octamer where the binding of proteins on one side of the nucleosome affects the dynamics of the other side of the nucleosome [13]. Recent NMR studies suggest the presence of dynamical networks within the nucleosomes where DNA dynamics are coupled to the dynamics of the globular core [14]. The above-mentioned facts highlight the dynamical complexity of the nucleosome and suggest that further research is needed. 

Among the arsenal of methods available to study nucleosome dynamics, molecular dynamics (MD) simulations have proven to be a useful tool, especially in combination with experimental methods such as NMR, cryo-EM, hydrogen-deuterium exchange, FRET, etc. All-atom MD simulations have been used to study histone tail dynamics [15,16], counter ion atmospheres [17], DNA wrapping/unwrapping [18,19,20], dynamics of sub-nucleosome particles [21], the effect of histone variants [22,23,24,25], histone post-translational modifications [26,27,28], DNA methylation [29], supra-nucleosome structures [30], etc. MD simulations can provide atomistic details about certain dynamical processes in nucleosomes and help to obtain mechanistic insights into the coupling between different dynamical modes. For example, in our recent study, we were able to observe a coupling between DNA sliding and plasticity of the histone octamer, which included changes in the conformation of the H2A-H2B dimer [20]. This prompted us to conduct a systematic analysis of H2A-H2B dimer dynamics in the present study. 

Since H2A-H2B dimers organize the segments of the nucleosomal DNA, which are close to the nucleosomal DNA ends, they are known to play important roles in modulating nucleosome dynamics, particularly DNA wrapping and unwrapping [31]. FRET studies suggested that DNA unwrapping may also be accompanied by the opening of the dimer–tetramer interface, the so-called butterfly states [32]. These dynamics are known to be modulated by histone sequence variants [33] and post-translational modifications [34,35,36,37]. Less is known about the internal dynamics of the H2A-H2B dimer, its plasticity and functional significance. Experiments on H2A-H2B cross-linking with glutaraldehyde have shown that cross-linked H2A-H2B dimers were only able to form hexosomes and unable to form complete nucleosomes [38], suggesting that H2A-H2B dynamics are needed for nucleosome assembly. Another interesting example is the ability of the SWR1 remodeler to distinguish between H2A- and H2A.Z-nucleosomes in yeast. This ability is significantly reduced if just two amino acid substitutions (corresponding to H2A.Z) are introduced into canonical H2A (G46K and P46A). These residues are not exposed on the surface of the nucleosome. It has been suggested that changes in the H2A-H2B dynamics and stability due to these substitutions are, in fact, responsible for these effects [39,40]. Overall, we expect that other dynamical modes of functional significance exist within H2A-H2B dimers awaiting characterization.
Figure 1Overview of the nucleosome core particle, H2A-H2B dimer structure, their elements and analyzed systems. (**a**) Nucleosome core particle (NCP). (**b**) H2A-H2B dimer. (**c**) H2A-H2B dimer bound to 30 DNA base pairs, H2A key arginines, which interact with the DNA minor groove on both sides of the H2A α2-helix, are shown in orange. Introduced reference frames (NRF—nucleosome reference frame, DRF—dimer reference frame) are shown on each panel. (**d**) H2B histone sequence and H2A-H2A.Z sequence alignment (*Xenopus laevis* canonical H2A and *Homo sapiens* H2A.Z). Basic and acidic residues are colored blue and red, respectively. Anchor arginines are highlighted by blue frames. Key structural elements are annotated below or on top of the sequences. Two positions on the alignment (highlighted in cyan frames) correspond to analog positions of H2A/H2A.Z substitutions in yeast affecting nucleosome stability and SWR1 activity [39,40]. ▼ and ▽ mark the histone tails’ truncation sites in NCP and dimer systems’ simulations, respectively.
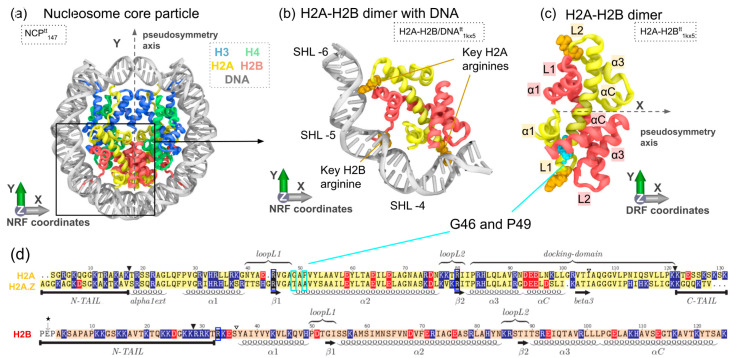


In the current study, we aimed to comprehensively characterize H2A-H2B dimers’ dynamics and plasticity using all-atom MD simulations and systematic analysis of available experimental structures. We performed a comparative analysis of H2A-H2B both in the free form and in the context of the nucleosomes. We specifically developed a framework for comparative structure analysis that makes use of the pseudosymmetry of the H2A-H2B dimer and nucleosome. Microsecond-long MD simulations allowed us to reveal the modes of dynamical plasticity of the globular core of the H2A-H2B dimer. The functional significance of H2A-H2B dynamics is discussed in relation to DNA sliding and wrapping/unwrapping. The effects of histone sequence on global and local dynamics of H2A-H2B dimers were demonstrated by simulations and analysis of the H2A.Z-H2B dimers. 

## 2. Materials and Methods

### 2.1. Atomistic Models Preparation for Molecular Dynamics (MD) Simulations

Atomistic systems were constructed based on X-ray nucleosome core particle (NCP) structures (PDB IDs 1KX5, 3LZ0, 1F66, see Appendix A). All systems (except H2A.Z-containing systems) contain *Xenopus laevis* histones and 601 Widom positioning sequences (3LZ0-derived structures) or α-satellite DNA (1KX5-derived structures). The systems had truncated flexible histone tails (Figure 1d). Truncated C- and N-termini of histones were capped by N-methyl (NME) and acetyl (ACE) non-charged groups. We built two H2A-H2B dimer models with the same sequence but slightly different atom coordinates, which were extracted from two X-ray nucleosome structures (1KX5 and 3LZ0). An H2A-H2B dimer with DNA was constructed entirely from a 1KX5 structure by selecting a 30 bp DNA segment (nucleotide numbers from −62 to −33 of chain I and from 33 to 62 of chain J) and H2A-H2B chains C and D. NCP with H2A.Z was constructed based on an X-ray structure with PDB ID 1F66 [41] and had *Homo sapiens* H2A.Z histone and *Xenopus laevis* H2B, H3 and H4 histones. Some histones lacked terminal residues, which were reconstructed using Modeller [42]. Using Chimera [43], the initial DNA sequence in the 1F66 structure was changed to the 601 positioning sequence to avoid DNA sequence-dependent specific effects. Models of nucleosomes with partially unwrapped DNA were constructed using the 3LZ0 reference structure and 3DNA package [44]. To this end, a segment of wrapped nucleosomal DNA was replaced by a straight B-DNA segment.

### 2.2. MD Simulations Details

Simulations were performed using established protocols described previously [20]. Briefly, we used GPU-accelerated GROMACS 2018.1 [45] with an AMBER ff14SB [46] force field supplemented with parmbsc1 [47] DNA and CUFIX [48] ion parameter corrections. Our models were placed in a truncated octahedron simulation box with periodic boundary conditions set at least 2 nm away from the dimer/NCP atoms. A TIP3P [49] water model was used for solvation, Na and Cl ions were added to neutralize the charge and bring the ionic strength to 150 mM. Then, the systems were minimized (the steepest descent gradient method for 10,000 steps with positional restraints on heavy atoms of 500 kJ mol^−1^ nm^−2^) and equilibrated in 5 steps (each 200-ps long) with decreasing positional restraints. MD simulations were performed in an NPT ensemble; the temperature was maintained at 300 K using a velocity rescale scheme [50] and pressure at 1 bar using a Parrinello–Rahman barostat [51]. The distances between terminal DNA nucleotides on each DNA end were constrained by a harmonic potential with a force constant of 1000 kJ mol^−1^ nm^−2^. This constraint was applied to all NCP models and an H2A-H2B dimer with DNA. Simulations were performed in parallel on the Lomonosov-2 supercomputer [52].

### 2.3. H2A-H2B Dimer Reference Frame Determination 

To compare the H2A-H2B dimer geometry in different systems (dimers, nucleosomes), a special coordinate reference frame (dimer reference frame, or DRF) was introduced. The DRF definition is based on the symmetry properties of the H2A-H2B dimer model derived from the 1KX5 X-ray structure. The X-axis was set to the second-order pseudosymmetry axis of the H2A-H2B and determined using CE-Symm v 2.2.0 [53]. The Y-axis was defined as an axis orthogonal to X-axis so that α2-helices Cα-atom projections onto this axis would achieve maximum mean absolute values. The Z-axis was defined as a cross-product of the X and Y axes. The calculations and dimer positioning in the DRF were performed in Python using Numpy [54] and MDAnalysis [55]. 

### 2.4. Analysis of MD Trajectories 

Custom analysis programs and pipelines were written in Python 3, integrating the functionality of GROMACS (trajectory preprocessing) [56], MDAnalysis (coordinate manipulation, 3D alignment, hydrogen bond analysis) [55], VMD and NGLview (visualization) [57,58], and 3DNA (determination of DNA base pair centers, calculation of base pair) [44].

The 1KX5-derived dimer was positioned in DRF, and then all other dimers and conformations were aligned to the 1KX5 dimer by minimizing the root-mean-square deviation (RMSD) between the Cα-atoms of the histone folds α-helices (α1, α2, α3). For NCP systems, the alignment to the nucleosome reference frame (NRF) for analysis as a whole system and in the DRF for dimer-oriented analysis was made. The definition of NRF was introduced in our previous article [20] and was based on the OY and OZ axes determination that corresponds to dyad and superhelical axes. The reference frame (NRF or DRF) is referenced in the figures. Any RMSD values reported in this study were calculated by first aligning the respective structures/conformations to the NRF or DRF and then computing RMSD without an additional round of pairwise alignment. 

The amplitudes of helices’ fluctuations were measured as the maximum distance between positions of the terminal Cα-atoms of the helices after dimer alignment to DRF. The angle between α2-helices (α2-α2 angle) was defined as the angle between vectors connecting the first and last Cα-atoms of the H2A and H2B α2-helices. The distributions (probability density functions) were visualized using a kernel density estimate with Gaussian kernels (SciPy realization). Hydrogen bonds were calculated using MDAnalysis, and the bond was defined with maximum donor-acceptor distance set to 3 Å and cut-off donor-hydrogen-acceptor angle set to 120°. The secondary structure elements were determined using the DSSP program [59]. DNA unwrapping in NCP systems was described as a length of unwrapped DNA ends (starting from either end of the nucleosomal DNA). A DNA segment is considered to be unwrapped from the NCP if the position of every base pair in that segment (as provided by 3DNA) was more than 7 Å away from the initial position of any base pair in the initial X-ray structure. 

### 2.5. Principal Component Analysis of H2A-H2B Dynamics 

To detect characteristic motions of H2A-H2B α2-helixes, we selected corresponding regions of histones from calculated MD trajectories and aligned them to the reference dimer structure. For two copies of dimer in the nucleosome, we obtained two aligned trajectories of helices for each copy and performed an independent analysis. Principal component analysis was applied to the collective notions of Cα-atoms (as implemented in Gromacs), and eigenvectors and corresponding eigenvalues were determined. A comparison of eigenvectors of each trajectory was made by calculations of the inner product of the eigenvectors; the inner product is 1 for codirectional vectors and 0 for orthogonal vectors. 

To compare the H2A-H2B bending dynamics characterized by the first eigenvector in independent trajectories, the projections of MD trajectories on this eigenvector from a 1KX5-based free dimer simulation were calculated with Gromacs. We chose a 1KX5-based free dimer simulation as a reference model since its first eigenvector was in good agreement with first eigenvectors in most other simulated systems (Appendix A). 

### 2.6. Structural Analysis of Experimental PDB Structures

Structures of H2A-H2B histone dimers with resolution greater than 4.0 Å were extracted from RCSB PDB by a sequence similarity search with canonical H2A and H2B histone sequences. Extracted structures were split into several groups: dimers belonging to single free NCPs (structures with 4 histone dimers and 2 DNA chains), dimers belonging to NCPs in complex with other proteins (4 histone dimers, 2 DNA chains and any other protein chains than histones), free dimers or dimers in complex with other non-histone proteins. Additionally, dimers from free NCPs were categorized by the DNA type based on the best match with 3 canonical sequences: Widom 601 sequence [60], α-satellite 146 bp long and α-satellite 147 bp long sequences [61] (other DNA sequences were excluded). All categories and their PDB codes are provided in Appendix A. All H2A-H2B dimers were extracted and aligned to the reference dimer positioned in DRF by minimizing the RMSD between the Cα-atoms of the histone folds α-helices (α1, α2, α3). Since structures contain slightly different sequences, chain names and residue numbers of all residues were mapped to the 1KX5-derived H2A-H2B dimer by pair sequence alignment. Root mean square variations (RMSV) were calculated for Cα-atoms of residues that occur in all structures. Resulting spatial variation metrics were compared between different structure groups (Figure 2f). Additional root mean square fluctuations of atomic positions (RMSF) values were estimated from the B-factors of Cα-atoms of best-resolved crystallographic structures of the NCP (1KX5) and H2A-H2B dimer (6K01) using the following formula RMSF = √(3B/8π^2^), where B is the B-factor value. Projections of atom positions were plotted to compare the ensemble of known structures. 

### 2.7. Interactive Materials 

Interactive materials were prepared using the NGL viewer JavaScript library and allow an interactive view of the MD trajectories, a visualization of PCA eigenvectors and projections of MD frames on the eigenvectors via interactive charts. The materials are hosted through a GitHub repository available at https://intbio.github.io/Kniazeva_et_al_2022 (accessed on 9 September 2022). 

## 3. Results

### 3.1. Overall Study Design and Research Methodology

We first revisited information available from the analysis of X-ray structures of the nucleosome core particle (NCP) [2], localization of H2A-H2B dimers in the context of the nucleosome, their structure and key interactions (see Figure 1). Each nucleosome consists of two copies of H2A-H2B and H3-H4 dimers, which are symmetrically arranged and spatially related via a two-fold pseudosymmetry axis passing through the center of the nucleosomal DNA (Figure 1a). In the context of the nucleosome, H2A-H2B dimers interact with the (H3-H4)2 tetramer (through the H2A docking domain and α3-helix of H2B) and with a nearby DNA region three turns in length (nucleotide numbers from 33 to 62 if counted from the dyad) (Figure 1b). Nucleosomal DNA interact with the octamer at 14 binding sites, each having a key arginine side chain inserted into the DNA minor groove. Each H2A-H2B dimer provides three DNA binding sites at superhelix locations (SHL) ±3.5, ±4.5, ±5.5: α1-α1 binding site in the middle of the dimer flanked by two L1L2 binding sites (Figure 1b). The key arginines at the L1L2 binding sites are provided by the H2A histone (H2A R42 and H2A R77, which both flank the long α2-helix of H2A), while at α1-α1 by H2B (H2B R33). The structure of the H2A-H2B dimer itself at its core is composed of two structurally similar histone fold domains, which interdigitate in a hand-shake manner forming a structure with a pseudosymmetric core (Figure 1c).

Previously, we have established a convenient reference frame, called the nucleosomal reference frame (NRF), in order to describe the conformation of the nucleosome and its functional motions [62]. The NRF is based on the nucleosome pseudosymmetry and superhelical axes. The introduction of NRF allows, in particular, to analyze 2D-projections of atomic coordinates and track changes within the nucleosome structure and relative positioning of its parts. Such an approach allowed us to track the motions of H2A-H2B dimers within NCPs during molecular dynamics (MD) simulations and show the association between DNA sliding and conformational changes within H2A-H2B dimers [63].

We aimed to introduce a similar approach to characterize the intrinsic dynamics of H2A-H2B dimers in a convenient reference frame. To this end, we determined the pseudosymmetry axis of the reference H2A-H2B dimer from the X-ray structure of NCP with the highest available resolution (PDB ID 1KX5 [61]) via CE-Symm v 2.2.0 [53] (see Methods). The second axis was defined as an axis orthogonal to the pseudosymmetry axis that maximizes the spatial variance of Cα-atom positions of histone α2-helices along this axis (see Methods), i.e., projections of α2-helices onto this axis will span the longest distance. The Z-axis was defined as a cross-product of the X- and Y-axes (see Figure 1c). These three axes establish the dimer reference frame (DRF), and the positioning of structures in the DRF allows to rationally analyze and describe conformational changes within the dimer. 

Next, our approach to studying H2A-H2B dynamics relied on comparative atomistic MD simulations. We took a reductionist approach to decipher the internal H2A-H2B dynamics and the effect of DNA and other histones in the context of the nucleosome on these dynamics. To this end, we simulated free H2A-H2B dimers, H2A-H2B dimers bound to 30-bp DNA segments and H2A-H2B dimers in the context of free NCPs. To probe the effects of different DNA sequences, systems based on ‘601’-Widom DNA sequences (3LZ0-derived structures) and the α-satellite DNA sequence (1KX5-derived structures) were used and comparatively analyzed. To probe the effects of DNA, the unwrapping of several NCP systems with initially unwrapped DNA was simulated. The full list of simulated systems can be found in Appendix A. All simulations were performed using all-atom representation in explicit solvent (150 mM NaCl solution). Flexible histone tails were truncated to speed up the sampling of conformational dynamics. The achieved length of MD simulations trajectories for dimer and free NCPs was no less than 3 μs (see Appendix A, interactive previews are available in Supplementary Interactive materials at https://intbio.github.io/Kniazeva_et_al_2022 (accessed on 9 September 2022)). Our analysis also included previously published [20] 8–15 μs NCP simulations. The analysis of dynamic modes was based on principle component analysis techniques as well as monitoring 2D-projection of atoms in the DRF.

To probe the influence of histone sequence on the dimer dynamics, we additionally simulated the systems containing *Homo sapiens* H2A.Z histone—H2A.Z-H2B dimer (3 μs) and H2A.Z-NCP (5 μs). We chose H2A.Z because this histone variant is one of the major histone variants found in all eukaryotes that has been extensively studied both in vivo and in vitro. Certain effects of H2A.Z on nucleosome stability and functioning have been biochemically mapped to specific amino acids; however, the exact dynamical mechanisms are not yet completely clear. For instance, in yeast, remodeler SWR1 was previously shown to distinguish the H2A-NCP from the H2A.Z-NCP, and the inner region (N-end of α2-helix) was shown to be critical for this discrimination [39,40].

As a complementary approach to MD simulations in order to analyze H2A-H2B dynamics and plasticity, we embarked on a systematic analysis of available PDB structures of H2A-H2B dimers, complexes of H2A-H2B dimers with other proteins, free nucleosomes and complexes of nucleosomes with other proteins. We have developed an automated approach to detect nucleosomes and H2A-H2B containing PDB structures, filter them according to the structure resolution, experimental method, and histone variants, align them into a common reference frame (DRF or NRF) and perform comparative analysis (see Methods). By setting the resolution threshold at 4 Å and retaining only canonical histones, the following structures were selected for analysis: 3 free H2A-H2B dimer structures (1 X-ray, 2 NMR), 19 X-ray and EM structures of the H2A-H2B dimer in complexes with other proteins, 157 nucleosome structures and 60 structures of the nucleosome in complexes with other proteins (see Appendix A).

How can the structural information from PDB be used to elucidate the H2A-H2B dynamics? While structures in PDB are usually reported in their averaged low energy conformation, we hypothesized that we could still extract some signatures of the dynamics from the following data analysis approaches. First of all, in traditional X-ray studies, the so-called B-factors for atoms are reported, which reflect their displacement due to thermal fluctuations. This displacement is likely in part due to the thermal fluctuations of the crystal lattice as a whole but should also include contributions from local thermal fluctuations due to the flexibility of H2A-H2B dimers. Another approach is to look at variations in H2A-H2B geometry in PDB structure (root mean square variation—RMSV). Likely, due to various conditions, differences in nucleosome composition, and interactions with other proteins, the conformation of H2A-H2B dimers will have to adapt to external constraints, and this adaptation will be mediated by the internal flexibility/plasticity of the dimer. As a third approach, ensembles of structures reported in NMR studies may be analyzed. We used all three approaches below to supplement our MD results and gain insights into the internal dynamics of H2A-H2B dimers.

### 3.2. MD and Experimental Evidence of H2A-H2B Histone Dimer Plasticity 

The combined analysis of H2A-H2B dimer dynamics and plasticity in MD simulations and experimental structures is presented in Figure 2 and Appendix A. MD simulations of free H2A-H2B dimers originally extracted from the NCP structure reveal that the overall structure of the H2A-H2B dimer globular core remains stable (note that we did not include flexible histone tails in simulations). However, certain elements become less structured due to the loss of contacts that are otherwise present in the context of nucleosomes. Such regions with RMSF of Cα-atoms of more than 2 Å include parts of the H2A docking domain starting with the α3-helix and N-terminal end of the α1 extension helix of H2A (see Figure 2a,f and Appendix A). L1L2 loop regions of H2A and H2B that form DNA-binding sites show somewhat higher dynamical fluctuations with respect to the α-helices in NCP simulations. Interestingly, in the free H2A-H2B dimers, the dynamics is increased asymmetrically with H2A L1 and H2B L2, showing higher fluctuations than H2A L2 and H2B L1 (see Figure 2f). The globular structure of the H2A-H2B dimer (operationally defined here as the positions of α1, α2, α3-helices) shows a certain extent of plasticity, which is higher for the free H2A-H2B dimer than for the NCP embedded one. The maximum RMSD value for the most different conformations of the globular core was 3 Å (see Appendix A) and 1.8 Å for the free and NCP embedded dimer, respectively. The positions of individual Cα-atoms of the H2A-H2B globular core could be displaced by around 7 Å during simulations. The analysis of 2D projections of Cα-atoms of α-helices in DRF (see Appendix A) shows increased dynamics of the C-end of the H2A α3-helix and increased dynamics of the C-ends of long α2-helices of H2A and H2B. The amplitude of α2 C-ends reaches 4.5 Å for the H2A helix and 6.8 Å for H2B in free dimers. For H2A-H2Bs embedded in the nucleosome, the amplitude of the C-end of the H2A α2-helix was estimated as 4.2 Å. However, if the fluctuations are analyzed within the nucleosome as a whole (i.e., we align individual snapshots not by superimposing individual dimer conformations but by the conformations of the nucleosome as a whole), the amplitude of the C-end of the H2A α2-helix is estimated as 7 Å. Considering both the internal plasticity of the dimer and dimer motion within the nucleosome as a whole may contribute to this effect. Interestingly, the dynamics of the α2-helices in the free H2A-H2B system are shifted towards the conformations of the helices in the direction away from the DNA binding surface (see Appendix A). This suggests the overall change in the shape of the H2A-H2B globular part upon its release from the nucleosome (see the next section for the detailed analysis of this bending mode). 

As an alternative approach to understanding dynamic fluctuations of H2A-H2B dimers, we extracted RMSF profiles from the reported B-factors in high-resolution X-ray structures of a nucleosome and H2A-H2B dimer, as well as analyzed the reported NMR structures ensembles of free H2A-H2B dimers (see Appendix A). The data of B-factors are in relatively good agreement with MD data, showing increased fluctuations in the L1L2 loop region of H2A-H2B and overall increased dynamics of the free dimer with respect to the NCP-embedded one. Although, the X-ray structure of the H2A-H2B dimer does not reveal disorganization of the H2A αC-helix. We also could not identify any crystal contacts that could contribute to this stabilization. The origin of this discrepancy remains to be investigated. The analysis of NMR data, however, suggests that the H2A αC-helix has some degree of conformational flexibility (see Appendix A). Contrary to the X-ray data, NMR data also suggest sufficiently increased mobility of the H2A α1-helix and H2B αC-helix (see Figure 2c and Appendix A). It is unlikely that such fluctuations would be possible in the context of an H2A-H2B crystal lattice where these helices participate in some contacts between neighboring monomers of H2A-H2B dimers. We did not observe fluctuations of these structural elements of such magnitude in MD simulations of free H2A-H2B dimers. This may be in part due to different ionic conditions used in MD simulations and NMR studies (150 mM NaCl vs. 400 mM KCl, respectively). 

The analysis of H2A-H2B structural variation in H2A-H2B complexes with other proteins, in different nucleosomes, and in complexes of nucleosomes with other proteins reveal interesting conformational variations that are in line with thermal fluctuations observed in MD simulations (see Figure 2f). H2A-H2B dimers show the least structural variation in the structures of free nucleosomes. The L1-L2 regions and the N-end of H2A αC-helix show increased variation similar to RMSF profiles of H2A-H2B dimers in MD. This structural variation may be in part due to different experimental conditions and different DNA sequences. The structural variation in nucleosome complexes shows a very similar pattern but is slightly higher, likely due to the structural perturbations caused by the interactors. The structural variation of H2A-H2B dimers interacting with different proteins outside of the nucleosome complex is the highest. Sufficiently increased variation within the L1-L2 loop regions, the H2A docking domain and H2B α3-αC helices are observed. Interestingly, the L2-loop region of H2B manifests a higher fluctuation than the L1-loop region, consistent with the structure fluctuations observed in MD. Some of the known structures in that group are the complexes of H2A-H2B dimers with chaperones, which are in good agreement with the free H2A-H2B dimer structure (Appendix A). An analysis of structural variation with 2D projections of the α-helices revealed that H2A-H2B plasticity within nucleosomes and nucleosome complexes is in line with the dynamics obtained in MD simulations (Appendix A). A detailed analysis of structural variation of H2A-H2B α2-helices in nucleosomes showed increased fluctuations of their ends, especially their C-ends (see Appendix A). The fluctuations were higher in the structures of nucleosomes interacting with proteins (see Figure 2f). 

### 3.3. H2A-H2B Bending Is a Major Dynamical Mode of Free and NCP-Embedded Dimer, Which Is Affected by Interactions with DNA and Other Histones 

Next, we aimed to systematically analyze the key collective dynamical modes of the H2A-H2B dimer globular core. To this end, we performed a principal component analysis of H2A-H2B dynamics in MD as represented by the motions of the two largest α-helices (H2A and H2B α2-helices). Such an approach allows one to focus only on motions engaging the whole H2A-H2B dimer without stochastic contributions from thermal fluctuations of small or disordered regions of the dimer, such as protein tails, loops, terminal helices, etc. A PCA analysis of the free H2A-H2B dynamics revealed that 35% of the structural variation is explained by the first two dynamical modes (see Figure 3a–c and Appendix A). The first mode explains around 24% of the dynamical variation and resembles simultaneous bending of the α2-helices so that the C-ends of the helices fluctuate along the direction of the dimer pseudosymmetry axis. The second dynamical mode also involves bending of the helices, but the direction of motion of the α2-helices’ C-ends lies in the perpendicular direction (see Figure 3c). We confirmed that the dynamical mode identified by the first eigenvector is also recovered with high accuracy in other MD simulations, including simulations of the H2A-H2B dimers embedded in nucleosomes (see Appendix A). 

We tried to introduce a geometrically defined collective variable that would independently quantify the conformation changes associated with collective motions described by the first eigenvector. Indeed, it turned out that the angle between the vectors connecting the ends of the α2-helices (the α2-α2 angle, see Figure 3b) correlates well with the projections of conformations onto the eigenvector (Pearson correlation coefficient −0.92). The distribution of conformations in MD trajectories was analyzed both in terms of α2-α2 angle values and eigenvector projection (see Figure 3d and Appendix A). The distributions for the free dimer, dimer embedded in NCP and dimer bound to DNA are broadly overlapping; however, the widths of the distributions were smaller for the NCP-embedded dimer (standard deviation 1.5 ° for NCP, 2.0 ° for free dimer and 2.1 ° for dimer bound to DNA) reflecting the constraints within the nucleosome. A clear shift in the maxima of the distribution is observed between the systems (see Appendix A). Particularly for the free H2A-H2B dimer, the most probable value of the α2-α2 angle is 1.6° lower than for the NCP embedded dimer. Interestingly, the most probable α2-α2 helical angle values for the H2A-H2B dimer bound by the DNA are almost identical to the NCP-embedded dimer. This suggests that the DNA pulls L1L2 binding sites closer together, deforming the dimer (see Figure 3e). This effect can also be seen in the analysis of 2D projections of α-helices when comparing different simulation systems (see Figure 3f,g). The system of the H2A-H2B with DNA was able to sample the largest values of the α2-α2 angle of around 154°, while the free dimer was able to sample the lowest values of the angle of around 138°. This gives us the potential span of α2-α2 helix motions accessible for the H2A-H2B dimer due to thermal fluctuations or interactions of more than 16°. 

Next, we aimed to analyze if, in the context of the nucleosome, DNA is the only force that contributes to the H2A-H2B dimer bending. To this end, we simulated an NCP with the DNA unwrapped from one end so that it interacted only with one binding site of the H2A-H2B dimer while other binding sites were free (NCP^tt^_50_ unwrapped). Surprisingly, the bending angle of the α2-α2 helices in this case was even larger than when the DNA is fully wrapped. This suggests that the histone octamer by itself has a certain influence on the H2A-H2B conformation. 

Previously, in the long-scale MD simulations of nucleosomes, we observed DNA sliding events that were accompanied by the inward bending of the C-end of the H2A α2-helix [20]. This bending weakens the interaction between DNA and histones at the L1L2 binding site and allows one of the DNA strands to slide past the site. Here we reanalyzed this motion in terms of intrinsic H2A-H2B bending. The originally observed bending in the context of NCP had two components: one associated with a shift of the dimer as a whole in the NCP structure and the second related to the intrinsic bending of the H2A-H2B dimer. The intrinsic bending of the dimer associated with DNA sliding measured in terms of the α2-α2 angle is estimated by us in this study as 142.9°, which is 3° off from the most probable value in the context of the NCP. Our analysis of the probability distribution suggests that the energetic cost of such intrinsic bending is around 2.5 kT (1.5 kcal/mol). 

### 3.4. Influence of DNA Sequence and DNA Unwrapping/Rewrapping on H2A-H2B Dimer Bending 

Nucleosomes are known to bind different DNA sequences. Nucleosome X-ray structures have revealed that depending on the DNA sequence, NCP can wrap 145, 146 or 147 DNA base pairs that would cover the same superhelical path. This is achieved by the capability of the nucleosome to include stretches of the DNA that are overtwisted and overstretched by one base pair. The known locations of such overtwisting (also called twist-defects) include SHL ±2 and ±5. The second location is within the DNA binding surface of H2A-H2B dimer. We hypothesize that DNA sequence may affect H2A-H2B dimer bending. The analysis of H2A-H2B dimer bending in X-ray structures harboring 145, 146, 147 DNA base pairs revealed that structures harboring 145 and 146 DNA base pairs had H2A-H2B dimers in slightly more bent conformations than the structures harboring 147 DNA base pairs (Appendix A). A similar tendency was observed during MD simulations of NCPs with different DNA sequences. NCPs harboring 147 DNA base pairs during MD simulations would relax into conformations with even lower α2-α2 angle values than observed in X-ray structures (Figure 4a and Appendix A). This relaxation usually happens within several nanoseconds and is likely due to the loss of crystallographic constraints by the system. However, in the case of NCP^tt^_146_ simulation (which is asymmetric and has DNA segments of different lengths on the two halves of the nucleosome), we observed a rather long relaxation process—it took around 4 microseconds of the system to reach stationary average values of α2-α2 angles (see Appendix A). 

The observed differences in H2A-H2B geometry are apparently conferred by the presence of the DNA twist-defect in NCP, harboring 145 DNA base pairs relative to NCP harboring 147 DNA base pairs. In this twist defect, the distance between the two L1L2 binding sites of H2A-H2B is spanned by a stretch of DNA that is one base pair shorter. Apparently, this generates the additional stress that tries to bring the binding sites together and bends the dimer. This idea is also supported by the following fact. In the NCP^tt^_145_ simulation, we observed the relaxation of the twist defect on one side. After this relaxation, the distribution of angle values of H2A-H2B changed to match the one observed in the NCP^tt^_147_ simulation (see Figure 4a–c).

Next, we decided to analyze the behavior of the H2A-H2B bending during the DNA unwrapping/rewrapping process. We have previously shown that DNA unwrapping/rewrapping is a multi-stage process, where the loss of stable interaction between DNA and histones is accompanied by rapid fluctuations of the DNA trying to re-establish the stable contacts [20]. We have prepared the system where contacts at the proximal L1L2 H2A-H2B binding site would be initially ruptured (NCP^tt^_25 unwrapped_). The rewrapping process took around 500 ns (Appendix A). The details of the rewrapping process are depicted in Figure 4d. It can be subdivided into two stages. During the first stage, the DNA end and the binding site try to form initial contacts. During this stage, substantial H2A-H2B dimer outward bending (towards the DNA) was observed, likely facilitating initial interactions of the binding site of the DNA. Once the DNA and the binding site come together, the arginine H2A R77 side chain starts to make contact with the phosphate backbone. During the next stage, the binding site and the DNA adapt to each other in order to establish stable contacts, including the insertion of the H2A R77 side chain into the DNA minor groove. Taken together, our observations suggest that the H2A-H2B bending facilitates DNA rewrapping. 

### 3.5. Histone Sequence Variants May Alter H2A-H2B Dimer Local and Global Dynamics: Example of H2A.Z Histone 

Next, we decided to examine the effects of the histone sequence on its dynamics. To this end, we constructed systems containing a *Homo sapiens* H2A.Z histone variant (see Methods). H2A.Z is one of the major histone variants that appeared early in eukaryotic evolution [64]. It plays important roles in many genomic processes, including heterochromatin regulation, DNA repair and transcriptional regulation (reviewed in [65]). We first examine the H2A-H2B bending mode identified earlier. The distribution of α2-α2 angles (see Figure 5a) and 2D projections of α2-helices (Appendix A) show interesting differences with respect to the dynamics of the canonical H2A-H2B dimer. For the free H2A.Z-H2B dimer, the maximum of the distribution is offset a little bit from the maximum of the canonical distribution towards the lower values of the α2-α2 angle. The distribution is also somewhat broader, reflecting higher flexibility of the H2A.Z-H2B dimer. In the context of the NCP, the distribution of α2-α2 angles is, on the contrary, quite substantially shifted towards the higher values of the α2-α2 angle (a shift of 3.8° is observed). During MD simulations of H2A.Z NCP, the DNA has undergone unwrapping/rewrapping from the proximal side of the NCP (Appendix A). However, even for that side, the distribution of the α2-α2 angle remained shifted toward higher values than for the canonical NCP. Overall, our observations suggest that the H2A.Z-H2B dimer has altered bending dynamics and is more susceptible to bending by external factors/forces. 

Recent experimental data for *S. cerevisiae* histones suggest that the differences between H2A and H2A.Z at two sites at the N-end of the α2-helix (G47K and P49A, which in humans correspond to G46T and P48A, see Figure 1d) are responsible for altered thermal stability of the nucleosomes and the capability for the SWR1 remodeler in yeast to distinguish between canonical and H2A.Z-containing nucleosomes [39,40]. Although H2A.Z in *S. cerevisiae* has a different amino acid substitution at one position (G47K in yeast vs. G46T in humans), H2A.Z histones across all eukaryotes have a sequence where glycine and proline at the N-end of the α2-helix in canonical H2A are substituted for different amino acids [66], highlighting the potential functional importance of these substitutions. We performed a detailed analysis of the conformational dynamics of the region near the beginning of the α2-helix. The analysis of secondary structure elements at the N-end of α2-helix revealed that it can undergo a transition between α-helical conformation and the conformation of the H-bonded turn (see Figure 5c,d). Such a transition was observed for both H2A and H2A.Z-containing systems in their free form or in the NCP-embedded form. However, the propensity to adopt various conformations varied substantially between H2A and H2A.Z systems. The equilibrium in the H2A.Z-containing systems was profoundly shifted towards the presence of the α-helical conformations near the T49 (G46 in H2A). This tendency was more significant for H2A.Z nucleosomes than for the free H2A.Z-H2B dimers. Structurally, the stabilization of the N-end of the α2-helix in H2A.Z is accompanied by the formation of the hydrogen bond between the carbonyl oxygen of residue A48 and backbone nitrogen of residue V52 (see Figure 5d). We observed that the loss of this hydrogen bond is accompanied by the formation of an H-bonded turn (confirmed by the Fisher exact test, *p*-value < 0.01), which maintains another hydrogen bond between the carbonyl oxygen of residue G47 and backbone nitrogen of residue A50 (see Figure 5d).

## 4. Discussion

In this study, we have performed an analysis of H2A-H2B dimer dynamics and structural variability using MD simulations and systematic analysis of structures from the PDB database. Free H2A-H2B dimers, dimers bound to DNA and dimers in the context of the nucleosome were simulated in all-atom representation at the timescale of several microseconds. PDB structures of free dimers, dimers interacting with different proteins, nucleosomes and nucleosomes interacting with different proteins were systematically analyzed. All core histones consist of the conserved histone fold motif (represented by three α-helices: -α1, α2, α3), decorated by additional α-helices, β-sheets and flexible histone tails. The dynamics of histone tails are highly disordered and were beyond the scope of our study, where we focused on describing the dynamics of the H2A-H2B elements that adopt a distinct structure in the nucleosome context. 

From MD simulations, we found that free H2A-H2B dimers demonstrate overall higher dynamics than in the context of nucleosomes. The dynamics are particularly increased for the H2A regions flanking the globular core (α1ext-helix, αC-helix and the docking domain) and the DNA binding sites formed by the L1-loop of H2A and L2-loop of H2B. The observed fluctuation profiles (RMSF) correspond well to the B-factors in X-ray structures, except for the H2A αC-helix, which does not show disorder in the X-ray structure of the H2A-H2B dimer. However, NMR studies including the hydrogen–deuterium exchange [67] suggest certain dynamics of the H2A αC-helix for the free H2A-H2B dimer. NMR studies of free H2A-H2B dimers in solution [68] also suggest sufficiently increased dynamics of the H2A α1-helix and H2B αC-helix that are not immediately observed in MD simulations nor in the X-ray structure of free H2A-H2B dimers. The reason for these discrepancies may be due to insufficient sampling time in MD simulations, higher salt concentrations used in NMR studies and crystal constraints limiting the dynamics of X-ray structures. Our analysis of the structural variation of H2A-H2B dimers in complexes with chromatin proteins supports the presence of higher dynamical variations for the H2A α1-helix and H2B αC-helix but also for the H2B α3-helix and both L1L2 loop regions. 

The compact histone fold core of the H2A-H2B dimer has also shown a considerable degree of conformational plasticity in MD simulations. The most different conformations had an RMSD of 3 Å, as measured for Cα-atoms of histone fold α-helices. The positions of individual Cα-atoms could be displaced by around 7 Å during the dynamics. PCA analysis revealed that a major contribution to these dynamics comes from the characteristic bending of C-ends of H2A and H2B α2-helices. This bending was further characterized in different contexts by measuring the angle between the helices and the distribution of these values. Changes of up to 16° have been observed. We have shown that upon binding to the DNA or incorporation into the nucleosome, the H2A-H2B dimer bends towards the DNA. The dimer bending in the nucleosome is also affected by the presence of DNA twist-defects. The DNA twist-defects localized in the nucleosome near the H2A-H2B dimer contribute to additional dimer bending when the DNA is overstretched between the H2A-H2B binding sites. We have previously observed that the relaxation of the DNA twist-defect through DNA sliding is accompanied by the inward movement of the L1L2 DNA binding site towards the center of the nucleosome [20]. This movement allows the phosphate group of the DNA backbone to slide past the DNA binding site. Here we have shown that this movement of the L1L2 binding site is due to simultaneous intrinsic bending of the H2A-H2B dimer and its slight shift within the nucleosome structure. We hypothesize that the intrinsic H2A-H2B bending facilitates nucleosome sliding. This has yet to be assessed experimentally through cross-linking of H2A-H2B at specific sites. However, experiments with disulfide cross-linking of H3-H4 dimers have previously shown that cross-links impede thermal nucleosome sliding [10], suggesting a common mechanism may exist for DNA sliding past H2A-H2B and H3-H4 dimers. 

We observed that during the DNA wrapping/unwrapping process, the H2A-H2B dimer also demonstrates bending dynamics, which apparently helps during the initial stages of DNA reattachment while bringing the binding site closer to the DNA. Interestingly, it has been shown experimentally that H2A-H2B dimers cross-linked by glutaraldehyde are unable to incorporate into a complete nucleosome particle [38]. 

We have also shown that the histone sequence may affect H2A-H2B dimer average bending and bendability. H2A.Z-H2B dimers were shown to be more flexible in the free form. In the nucleosome-embedded form, we show considerably higher bending towards the DNA. These observations are in line with recent structural studies of other histone variants. Particularly, Sato et al. [69] have shown that nucleosomes based on histones from G. lamblia demonstrated outward bending of the H2A α2-helix. Concomitantly, Zhou et al. [33] observed that H2A.B-containing nucleosomes had a 15° kink in the C-terminal region of the H2A.B α2-helix, although this kink was more in the direction along the nucleosomal superhelical axis. Such motion would correspond to the second eigenvector identified in our PCA analysis of a2-helices’ dynamics. Figure 6 summarizes the potential functional implications of H2A-H2B bending.

By the example of H2A.Z-H2B dynamics analysis, we have shown that histone sequence also may affect the local dynamics of histone fold elements. Dai et al. [40] have previously shown that substitutions of G47K and P49A in yeast H2A.Z with respect to canonical H2A increase dimer and nucleosome stability, which in turn contributes to the ability of SWR1 to discriminate between H2A and H2A.Z nucleosomes [39]. Similar amino acid substitutions of glycine and proline at the N-end of the α2-helix are found in H2A.Z across eukaryotes, including human H2A.Z that we have studied. Here we have shown the dynamical basis for the potential increase in the stability of H2A.Z nucleosomes. Namely, the N-end of α2-helix was found to be in dynamical balance between α-helical conformation and conformation of the H-bonded turn. The amino acid substitutions found in H2A.Z considerably stabilize the α-helical conformation, which likely helps to form more stable contacts with the DNA. 

Taken together, in the present study, we have provided a comprehensive analysis of dynamics and structural variation of an H2A-H2B histone dimer globular core using MD simulations and systematic analysis of PDB structures. We have identified and characterized new dynamical modes, including bending of the histone α2-helices. We have shown in simulations that the intrinsic plasticity of the H2A-H2B dimers provided by these modes has functional implications. Particularly, the bending of α2-helices facilitates the sliding of DNA phosphate groups at the histone-DNA binding sites, which likely facilitates further nucleosome sliding as a whole along the DNA. H2A-H2B plasticity was also shown to affect kinetics of DNA rewrapping. Given the vast repertoire of histone post-translational modifications and histone sequence variants used in genome functioning, we hypothesize that their effects on histone dimer plasticity may be one of the mechanisms in regulating chromatin structure and dynamics. Previously, attention has been focused mainly on histone PTMs located at histone-DNA or histone dimer–dimer interfaces, which have been suggested to affect nucleosome stability [70,71]. However, a few PTMs are also known to be located within the globular core of the H2A-H2B dimer. For instance, it has been shown that H2AY57ph decreases dimer thermostability [36] and thus likely increases its dynamics.

Our study is not without limitations. Particularly, our MD analysis was limited by the timescale of several microseconds. It is likely that at a longer timescales, other dynamical modes of higher magnitude within the H2A-H2B dimer may manifest. Another limitation in our current analysis is the absence of flexible histone tails in simulations. The dynamics of the histone tails likely affect the dynamics of dimers and nucleosomes at longer timescales. 

## Figures and Tables

**Figure 2 cells-11-02837-f002:**
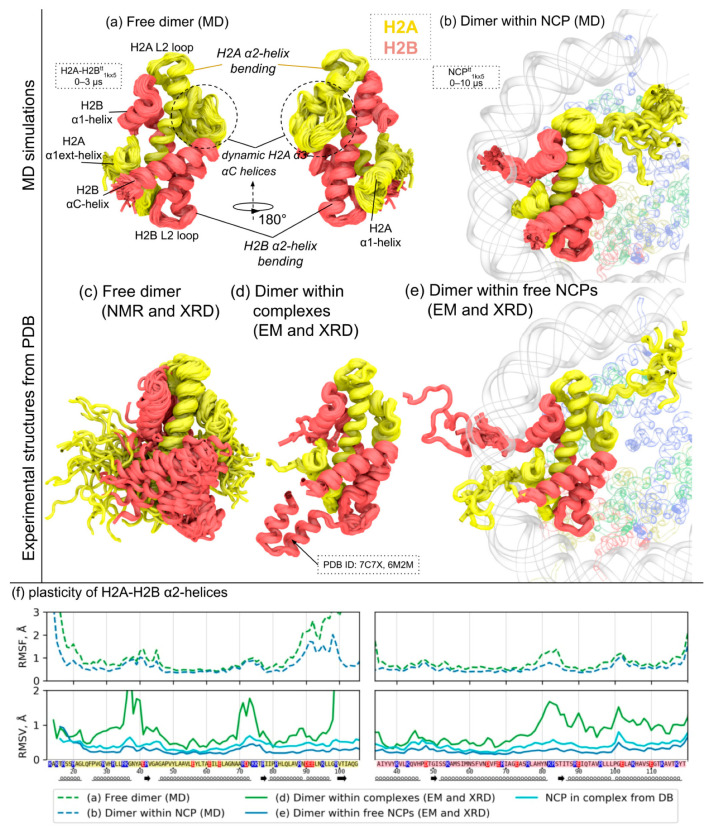
H2A-H2B dimer dynamics in molecular dynamics (MD) simulations and experimental data. (**a**,**b**) Overview of H2A-H2B dimer dynamics in MD simulations as a free dimer (**a**) or in the context of NCP (**b**). MD snapshots spaced 10 ns apart are overlaid. The respective IDs of simulated systems are depicted in dashed boxes (see Appendix A for system description). (**c**–**e**) Overview of structural variation in H2A-H2B dimers observed in PDB, grouped by structure type and experimental origin: (**c**) superimposed structures of free H2A-H2B dimers resolved via X-ray crystallography (XRD) and structural models from NMR studies, (**d**) superimposed structures of H2A-H2B dimers from complexes with other proteins (except nucleosomes), (**e**) superimposed structures of H2A-H2B dimers in the context of free NCPs found in PDB. PDB IDs of depicted structures can be found in Appendix A. (**f**) Analysis of structural fluctuations and variations for Cα-atoms of H2A (**left**) and H2B (**right**) histones. The plots show the values of either root-mean square fluctuations (RMSF) from MD or root-mean square variations (RMSV) of atomic positions between PDB structures of a given group (see legend). Each group of experimental structures or MD simulations is represented by a separate line according to the legend.

**Figure 3 cells-11-02837-f003:**
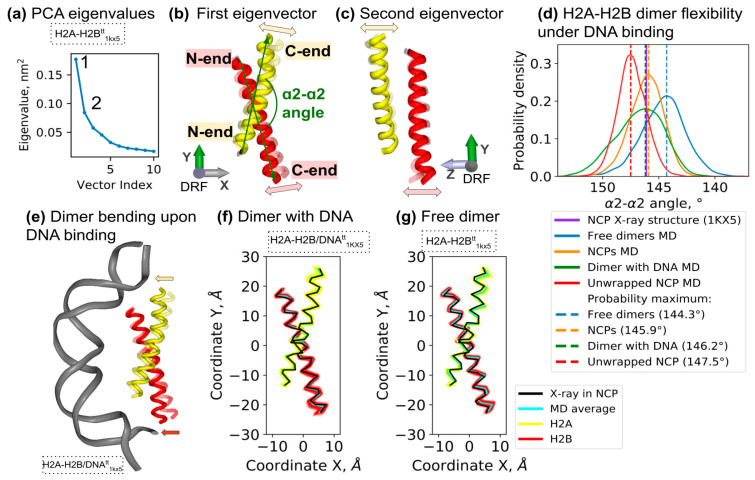
Intrinsic H2A-H2B dimer bending and effects of interaction with DNA. (**a**–**c**) Principal component analysis of histone α2-helices’ Cα-atoms thermal fluctuations in MD. (**a**) Eigenvalues of the first 10 eigenvectors. (**b**,**c**) Dynamical modes corresponding to first and second eigenvectors. Snapshots of two extreme states along the eigenvectors are shown (see Appendix A and interactive materials for a more detailed view). The angle between the two α2-helices is shown in (**b**) (see Methods for definitions). (**d**) Probability distributions of α2-α2 angle values during MD simulations of different systems: free H2A-H2B dimers (systems H2A-H2B^tt^_1KX5_ and H2A-H2B^tt^_3LZ0_), dimer bound to DNA (H2A-H2B/DNA^tt^_1KX5_) and NCPs (averaged from NCP_147_, NCP^tt^_147_, NCP^tt^_146_, NCP^tt^_145_). The dashed vertical lines mark the maximum distributions, and the solid violet vertical line marks the value in 1KX5 X-ray NCP structure. The corresponding plots for all simulated systems are shown in Appendix A. For statistical analysis, see Appendix A. (**e**) Schematic representation of H2A-H2B bending upon DNA binding. Arrows mark the direction of the bending. (**f**,**g**) Two-dimensional projections of histone α2-helices conformations in MD of free (**f**) and DNA-bound dimer (**g**). The mean helix conformations and conformations in an NCP X-ray structure are shown.

**Figure 4 cells-11-02837-f004:**
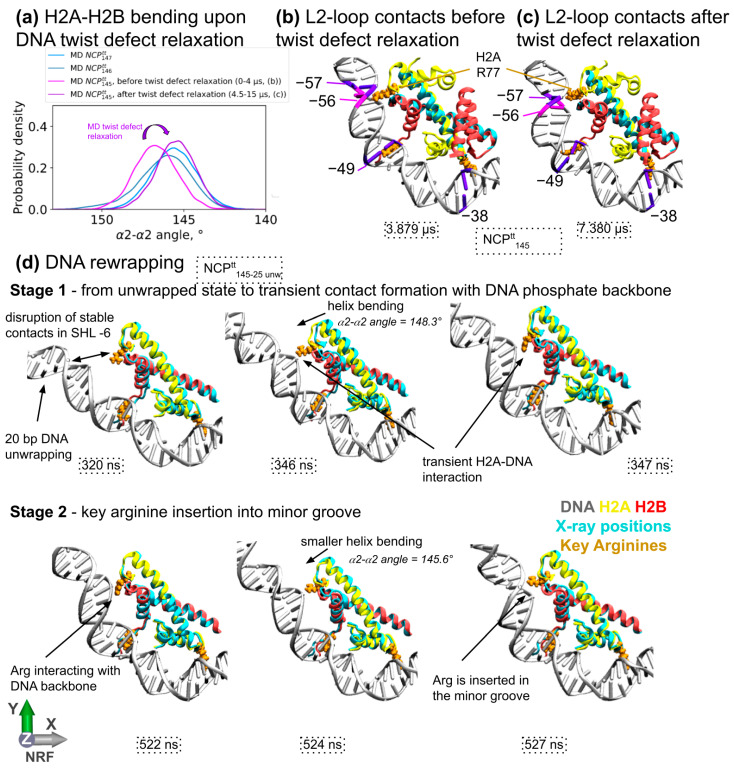
Interplay between H2A-H2B dimer bending, DNA twist-defects and wrapping/unwrapping dynamics. (**a**) α2-α2 angle distributions for different simulated NCP systems. (**b**,**c**) Snapshots illustrating DNA geometry and H2A-H2B bending upon twist-defect relaxation in the NCP^tt^_145_ system. (**d**) MD snapshots illustrating the DNA rewrapping process and the H2A-H2B bending during this process.

**Figure 5 cells-11-02837-f005:**
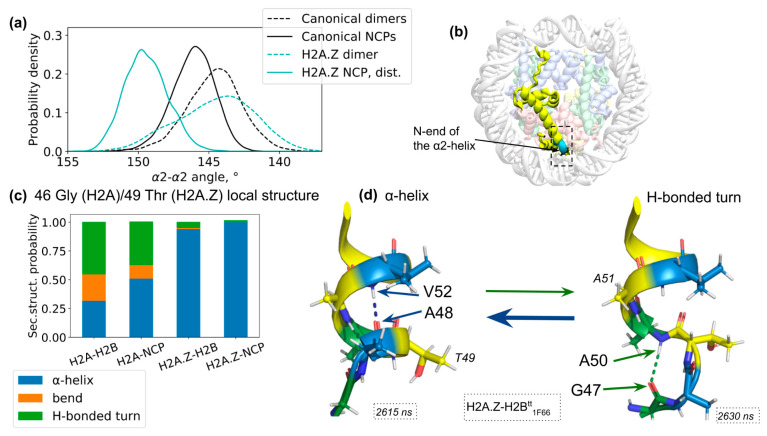
Effects of H2A.Z histone incorporation on the H2A.Z-H2B dimer and NCP dynamics. (**a**) α2-α2 angle distributions for canonical H2A-H2B and variant H2A.Z-H2B dimers free in solution and in the context of nucleosomes. (**b**) Location structural elements of H2A.Z histone in the context of nucleosome with particular emphasis on the changes at the N-end of the α2-helix (substitutions of G46 and P48 in H2A to T49 and A51 in H2A.Z). (**c**) Probability of the N-end of the α2-helix of H2A participating in various secondary structure elements depending on the histone variant (H2A/H2A.Z) or presence of the nucleosome context. The secondary structure analysis was obtained by DSSP for residue G46 or T49 for H2A or H2A.Z, respectively (see more details in Appendix A). (**d**) Illustration of structural transition between a-helix and H-bonded turn in the N-end of α2-helix of H2A/H2A.Z.

**Figure 6 cells-11-02837-f006:**
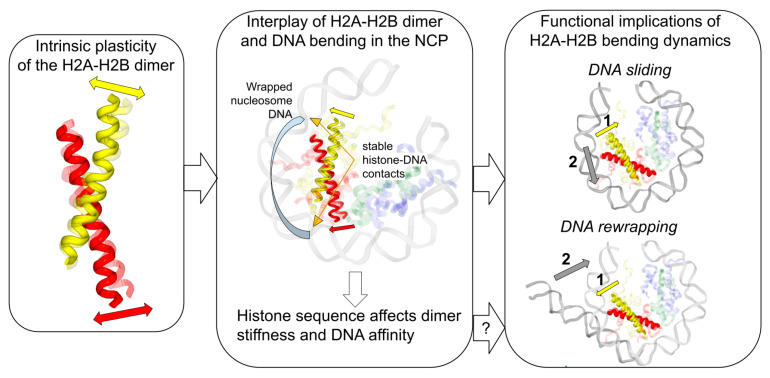
Functional implications of H2A-H2B dimer plasticity.

## Data Availability

Trajectories and protocols are available for preview and download from GitHub at https://intbio.github.io/Kniazeva_et_al_2022 (accessed on 9 September 2022).

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
