# Peer review of "H2A-H2B Histone Dimer Plasticity and Its Functional Implications"

_cells, 2022, doi:10.3390/cells11182837_

Round 1

Reviewer 1 Report

The paper on H2A-H2B Histone Dimer Plasticity and its Functional Implications is well written, and really interesting.

The study has been thoroughly conducted and provide all needed information. 

Minor: 

I would suggest to add the limitations of the study in the discussion section and I would recommend to clearly write the functional implication of these findings as well as some perspectives.

Reviewer 2 Report

Kniazeva et al. have tried to elucidate the H2A-H2B dependent mechanisms using X-ray crystal structures via simulations and docking approaches.

The study is quite interesting. However, I have the following suggestions and comments.

1) Because PTMs play a significant role, it is unclear how PTMs affect proposed mechanisms. 

2) How the plasticity affects higher-order chromatin structures, and does it affect gene regulation?

3) How this plasticity is modulated by histone chaperones and affected in different cellular conditions.

Author Response

Response to reviewers’ comments

RE: Manuscript “H2A-H2B Histone Dimer Plasticity and its Functional Implications” by Kniazeva et al.

Original comments are highlighted in bold blue, authors’ answers in black normal font, changes to the manuscript in “green italics”.

Reviewer #2.

Kniazeva et al. have tried to elucidate the H2A-H2B dependent mechanisms using X-ray crystal structures via simulations and docking approaches. The study is quite interesting. However, I have the following suggestions and comments.

We thank the referee for positive feedback on our manuscript.

1) Because PTMs play a significant role, it is unclear how PTMs affect proposed mechanisms.

We thank the referee for raising this question. As we show histone globular core plasticity can be altered by sequence alterations, and PTMs are expected to modulate dimer plasticity. Talking particularly about the bending mode we should note that rare PTMs are present in vivo in the H2A-H2B a2-helices. One example of such PTMs is H2A Tyr57 phosphorylation in the H2A a2-helix, which has an effect on the dimer thermostability, another example is H2BK57ac.

We added a short discussion of this issue at the end of the manuscript.

“Given the vast repertoire of histone post-translational modifications and histone sequence variants used in genome functioning we hypothesize that their effects of histone dimer plasticity may be one of the mechanisms in regulating chromatin structure and dynamics. Previously, attention has been focused mainly on histone PTMs located at histone-DNA or histone dimer-dimer interfaces which have been suggested to affect nucleosome stability [70,71]. However, few PTMs are also known to be located within the globular core of the H2A-H2B dimer. For instance, it has been shown that H2AY57ph decreases dimer thermostability [36], and thus likely increases its dynamics.”

2) How the plasticity affects higher-order chromatin structures, and does it affect gene regulation?

Currently, we can only speculate that plasticity may be involved in gene regulation or higher-order chromatin structuring through facilitating or affecting nucleosome sliding. For instance, certain potential mutations that increase plasticity will likely make nucleosomes less prone to hold their positions in a particular genome locus.

3) How this plasticity is modulated by histone chaperones and affected in different cellular conditions.

We thank the referee for bringing this to our attention. There are only a few structures with canonical H2A-H2B dimer in the complex with chaperones; we added its overlay with free H2A-H2B dimer structure (see Figure S4b) and corresponding description in the 3.2 section. However, we did not observe significant rearranges in dimer bending upon chaperone binding.

We added the following text to the results.

“Some of the known structures in that group are the complexes of H2A-H2B dimer with chaperones, which are in a good agreement with the free H2A-H2B dimer structure (Figure S4b).”

Reviewer 3 Report

This is an interesting work focusing on evaluating computationally the detailed connection between the sequence of H2A-H2B histones, their structure, dynamics and its implications for nucleosome functioning.
Several sections of the results are too verbose and should be shortened for a better reading. Despite reporting numerous data, the main limitation of the study is the lack of experimental validation of the reported findings. Figures are highly informative, detailed and clear. My recommendation is minor revision. Heres several comments for improving the work.
a)    The species from which histones were computationally analyzed should be detailed in the abstract.
b)    A couple of introductive words on the main histone post-translational modifications and how they modulate the DNA wrapping and unwrapping would be helpful for the reader. Authors can check and include PMID: 25424540 and PMID: 29548294
c)    The aim of the study at the end of the introduction can be shortened for a better reading
d)    The discussion should be more deeply detailed and strengths and weaknesses included, for instance, the findings of the work should be validated experimentally. A couple of words can be included on this issue and further perspectives
e)    Line 46 Histone post-translational modifications can be considered as therapeutic targets for tumor therapy (PMID: 35350569). For completeness, this information and supporting reference should be  introduced.
f)    Spaces between paragraphs in the same section can be removed, for instance lines 134, 154, 186, 606, 623 etc. Please revise the text of other unnecessary spaces
g)    The results are too verbose and can be shortened by 20% for a better reading
h)    Line 350 “â„« for” please revise this typo
